# The Role of Collagen VIII in the Aging Mouse Kidney

**DOI:** 10.3390/ijms25094805

**Published:** 2024-04-28

**Authors:** Ngoc Dong Nhi Vo, Nikolaus Gaßler, Gunter Wolf, Ivonne Loeffler

**Affiliations:** 1Department of Internal Medicine III, University Hospital Jena, 07745 Jena, Germany; dongnhi.vo@med.uni-jena.de (N.D.N.V.); gunter.wolf@med.uni-jena.de (G.W.); 2Institute of Forensic Medicine, Section Pathology, University Hospital Jena, 07745 Jena, Germany; nikolaus.gassler@med.uni-jena.de

**Keywords:** collagen VIII, aging kidney, fibrosis, sex differences

## Abstract

The gradual loss of kidney function due to increasing age is accompanied by structural changes such as fibrosis of the tissue. The underlying molecular mechanisms are complex, but not yet fully understood. Non-fibrillar collagen type VIII (COL8) could be a potential factor in the fibrosis processes of the aging kidney. A pathophysiological significance of COL8 has already been demonstrated in the context of diabetic kidney disease, with studies showing that it directly influences both the development and progression of renal fibrosis occurring. The aim of this study was to investigate whether COL8 impacts age-related micro-anatomical and functional changes in a mouse model. The kidneys of wild-type (*Col8*-wt) and COL8-knockout (*Col8*-ko) mice of different age and sex were characterized with regard to the expression of molecular fibrosis markers, the development of nephrosclerosis and renal function. The age-dependent regulation of COL8 mRNA expression in the wild-type revealed sex-dependent effects that were not observed with collagen IV (COL4). Histochemical staining and protein analysis of profibrotic cytokines TGF-β1 (transforming growth factor) and CTGF (connective tissue growth factor) in mouse kidneys showed significant age effects as well as interactions of the factors age, sex and *Col8* genotype. There were also significant age and *Col8* genotype effects in the renal function data analyzed by urinary cystatin C. In summary, the present study shows, for the first time, that COL8 is regulated in an age- and sex-dependent manner in the mouse kidney and that the expression of COL8 influences the severity of age-induced renal fibrosis and function.

## 1. Introduction

Life expectancy has significantly increased in recent decades, while mortality rates have decreased, mainly in the Western world, but also in other countries. This has resulted in the rapid aging of the world’s population [1]. The increasing numbers of elderly individuals are a burden for the health systems, because the diagnoses of age-related limitations are also increasing.

Aging is a natural and complex biological process that leads to the gradual deterioration of various organ systems, including the kidneys [2]. Renal aging is characterized by clinical and functional changes, which are accompanied by structural changes [3]. Systemic comorbidities such as hypertension, diabetes mellitus, or pre-existing renal disease may contribute to or exacerbate these changes [4,5]. Besides genetics, race, oxidative stress, angiotensin II, impairment in kidney repair ability and chronic inflammation, gender/sex is also an important factor that is associated with renal aging [6]. The micro-anatomical changes that occur with aging include glomerulosclerosis, pericapsular fibrosis, arteriosclerosis, tubular atrophy and interstitial fibrosis [3]. In a study of kidney transplant donors, nephrosclerosis, defined as the combination of two or more of glomerulosclerosis, tubular atrophy, interstitial fibrosis, or arteriosclerosis, increased linearly from 2.7 percent in the kidneys of the young (18- to 29-year-olds) to 73 percent for ages 70 to 77 y [7,8]. The overall number of functional tubules decreases with age, and tubular length and volume are also markedly reduced [6]. Tubulointerstitial fibrosis in aging kidneys is characterized by excessive collagen deposition and structural changes in the extracellular matrix, the altered regulation of the expression of metalloproteinases and TGF-β1, and the activation of fibrosis- and hypoxia-related genes [6].

In addition to other collagens, such as collagen IV (COL4; the major constituent of basement membranes and the specialized form of the extracellular matrix), collagen VIII (gene: *Col8*; protein: COL8) is also robustly expressed in the kidney. Just as COL4 underlies the pathogenesis of various disorders, including diabetic nephropathy (DN), we have found a specific induction of COL8A1 expression in glomerular and tubular cells in murine experimental DN [9].

First described as endothelial collagen by Sage et al. in 1983, COL8 is a non-fibrillar collagen with short chains and the ability to form complex hexagonal lattice structures. It is synthesized by a variety of cell types (endothelial cells, smooth muscle cells, mast cells, monocytes, macrophages and T cells) and is prominently deposited at various highly specialized matrices, such as vascular walls and the Descemet membrane of the eye [10,11,12]. Two genes coding for different polypeptide chains, α1(VIII) and α2(VIII), have been identified that can form both heterotrimeric and homotrimeric molecules, and thus, four different triple-helical COL8 proteins can exist in vivo [11,13,14]. In particular, COL8A1 is expressed by all resident glomerular cell types (mesangial, endothelial and epithelial cells) as well as by cells from the tubular compartment (e.g., tubule cells) [9,15].

Although COL8 is structurally well characterized, its biological function or how COL8 interacts with other ECM components or the basement membrane structures is largely unclear. It has been suggested that this network-forming collagen may act as a “FACIT” (Fibril Associated Collagens with Interrupted Triple helices)-like molecule and polymerize or link other matrix molecules [12]. COL8 is also thought to activate the receptor DDR1 as an agonist and to transmit signals into cells via integrin β1 [16,17,18]. However, a direct relationship between the elevated COL8 expression and the induction of profibrotic cytokine TGF-β has been demonstrated through cell culture experiments in that it has been shown that TGF-β can induce the expression of COL8 in mesangial and tubular cells [9]. A renoprotective effect showed the knockout of *Col8* (*Col8*-ko) in diabetic mice [9,19]. Compared with the diabetic wild-type mice, the diabetic *Col8*-ko mice showed significantly ameliorated albuminuria, which resulted from reduced glomerular changes [9]. Further investigations into the role of COL8 in the tubulointerstitial fibrosis of DN confirmed that the knockout of COL8 is renoprotective, and therefore, significantly reduced fibrosis in the tubular compartment was found [19].

This study investigates, for the first time, renal changes depending on the three factors, age, sex and *Col8* genotype, with a focus on fibrotic tubulointerstitial changes in a mouse model.

## 2. Results

### 2.1. Age- and Sex-Dependent Regulation of Renal Col8a1, Col8a2 and Col4a1 Expression

To determine if there is an age-dependent regulation of collagen VIII, RT-PCRs were performed on tissue samples from the renal cortex. Male and female wild-type mouse kidneys (*Col8*-wt) of different ages were used to record *Col8a1* and *Col8a2* mRNA expression throughout the lifespan of the mouse. The study included five age groups: very young (6–12 weeks), very young to young (12–24 weeks), middle-aged (24–40 weeks), middle-aged to old (40–72 weeks), and old (72–100 weeks). The results showed a significant increase in *Col8a1* and *Col8a2* mRNA expression in male animals as they aged, with the most significant difference observed between the oldest male group and the other age groups (Figure 1a,b). The expression pattern in females differed from that in male; with regard to *Col8a1*, it exhibited a peak in mRNA abundance in the middle-aged group and remained relatively constant until old age. In comparison, the mRNA expression of *Col8a2* peaked at a young age, declined, and then increased again in middle age before decreasing in old age. A multifactorial ANOVA with the factors sex and age revealed a significant sex (*p* = 0.008) and biologically relevant age (*p* = 0.081) effect on *Col8a1* regulation and a significant interaction between sex and age (*p* < 0.001) in the regulation of *Col8a2.* To preview future investigations into fibrosis, the same samples were analyzed for the mRNA expression of collagen IV alpha 1 (*Col4a1*), the major component of renal basement membranes (Figure 1c). It has been observed that the expression and progression of *Col4a1* and the *Col8* chains are similar in males over time, but this was much less pronounced than for *Col8a1* and *Col8a2.* In the female sex, there was a decrease in expression in the older groups compared to the youngest group, which, as with *Col8a2*, was significantly higher than the equivalent male age group. In contrast to *Col8* chains, for *Col4a1,* multifactorial ANOVA analysis did not reveal any significant or biologically relevant interactions.

### 2.2. Influence of COL8 on Fibrotic Changes in the Aging Kidney

For our studies on the influence of COL8 on the age-related structural and functional changes of the kidney, we used mice at an average age of 21.4 weeks (12–25 weeks, referred to as young) and 85.1 weeks (73–93 weeks, referred to as old). The age groups used correspond in humans to an age of 20–30 years, also referred to as mature adults, and an age of 56–69 years, defined as old [20]. According to another calculation formula, which assumes that 9 mouse days (of mature adults) correspond to one human year [21], the translated human age of our old groups used is 68–82 years.

To investigate the effects of the absence of COL8 protein in all its four possible triple-helix variants on the aging kidney, we used genetically modified double-knockout animals, i.e., animals in which both the *Col8a1* and *Col8a2* genes were knocked out (*Col8*-ko).

Structural fibrotic changes, as a consequence of the altered expression of molecular fibrosis mediators, can be visualized with Masson staining based on the blue staining of fibrotic tissue. Both glomerular and tubulointerstitial evaluation indicated a clear age-, COL8-, and sex-dependent effect (Figure 2a,b). Starting with glomerular staining (Figure 2a), there was a tendency for increased fibrosis in older male wild-types compared to younger ones. The *Col8*-ko resulted in a higher degree of fibrosis in aged kidneys, which was significant compared to the young knockout animals. There were neither biologically relevant nor statistically significant changes in the degree of fibrosis between young and old female wild-types. However, in the *Col8*-ko females, there was a notable development of fibrosis at an advanced age. In other words, in the old *Col8*-wt of both sexes, a difference in the degree of fibrosis was noticeable, with fibrosis formation being more pronounced in old male wild-types than in old female wild-types. The effect was nullified by the *Col8*-ko, making it impossible to determine any difference. The multifactorial ANOVA revealed significant effects, including an age effect (*p* < 0.001), interactions between the *Col8* genotype and age (*p* = 0.007), and a biologically relevant three-way interaction between the *Col8* genotype, sex, and age (*p* = 0.091).

The evaluation of tubulointerstitial staining yielded comparable results for both sexes (Figure 2b). In the male *Col8*-wt and *Col8*-ko mice of both age groups, no significant changes were observed. However, in kidneys from female mice, *Col8*-ko had again a significant effect. Whereas no significant differences were found between young and old female wild-types, the degree of fibrosis increased significantly in old *Col8*-ko females compared to the old wild-type group and the young knockout group of the same sex, as previously demonstrated by the evaluation of glomerular staining. The multifactorial ANOVA revealed significant effects, including significant age effects (*p* < 0.001) and a significant three-way interaction between the *Col8* genotype, sex, and age (*p* = 0.017).

### 2.3. COL8 Amplified Age-Dependent Tubular Dysfunction

It has been reported that urinary cystatin C concentration increases with renal tubular damage, independent of change in GFR, and that the cystatin C/creatinine ratio may be a suitable non-invasive method to detect renal dysfunction [22,23]. Therefore, urinary measurements of cystatin C (UcysC) and creatinine (Ucrea) were performed and the ratio of both was calculated to determine renal tubular function in our mouse model (Figure 3). A slight increase in cystatin C excretion in the old wild-type group of both sexes was detectable when comparing to the corresponding young groups (Figure 3a). The *Col8*-ko further exacerbated the age-dependent increase in cystatin C excretion, which could be observed in males as well as females. The multifactorial ANOVA revealed a significant effect of the *Col8* genotype (*p* = 0.006) and age (*p* < 0.001), and a significant interaction between the *Col8* genotype and age (*p* = 0.014) across all groups. The results of the urinary creatinine measurement were very similar to the results of UcysC with enhanced age-dependent excretion in the *Col8*-ko, which was also reflected in a highly significant genotype–age interaction (*p* = 0.002) (Figure 3b). The differences observed in the individual measurements were also apparent when calculating the UcysC/Ucrea ratio (Figure 3c). Multifactorial ANOVA revealed again a significant effect of the *Col8* genotype (*p* = 0.043) and a significant age effect (*p* = 0.003).

### 2.4. Col8 Knockout Disrupted the Physiological Architecture of the Renal Matrix

COL4A1 is, as already mentioned, the main component of the glomerular and tubular basement membranes, but also of the mesangial matrix and Bowman’s capsule. It is known that the expression of COL4 is increased in renal fibrosis, leading to mesangial sclerosis and the thickening of the glomerular and tubular basement membranes [24]. As shown in Figure 4, the age-dependent increase in COL4A1 expression in wild-type males and decrease in wild-type females, as observed in the mRNA analyses (see Figure 1c), could not be confirmed with immunohistochemical staining. Age had no effect on renal COL4A1 protein expression in male animals, but it led to a decrease in expression in wild-type female kidneys. Furthermore, the results for COL4 stainings of the kidneys from *Col8*-ko mice were very surprising; namely, the absence of COL8 resulted in a significant decrease in the expression of COL4A1 in both sexes and age groups. The effect of the knockout was highly significant when comparing the young *Col8*-ko groups with the young wild-type groups of both sexes. The three-way ANOVA also revealed a highly significant *Col8* genotype effect (*p* < 0.001). While in old wild-type kidneys compared to young ones were no changes respectively a reduction of COL4A1, the old *Col8*-ko kidneys showed an increase in COL4 staining compared to the kidneys from the young animals of the same genotype, which is reflected in a significant interaction between *Col8* genotype and age (*p* = 0.008) in the ANOVA analysis

### 2.5. Analysis of Profibrotic Marker Expression in the Aging Kidney Depending on Col8 and Sex

Age-related fibrotic molecular changes in the kidney and their dependence on sex and COL8 were finally analyzed by measuring the key mediator of profibrotic pathways and tissue fibrosis TGF-β1 (transforming growth factor β1) by ELISA with tissue homogenates from the renal cortex (Figure 5a). In fact, the TGF-β1 expression increased with age in both *Col8*-wt and *Col8*-ko mice of both sexes (significant age effect (*p* = 0.018) assessed by ANOVA analysis). Based on the median in the intergroup comparisons, the TGF-β1 levels in female kidney lysates (except in the young wild-type group) were slightly lower than in the corresponding males, which is even biologically relevant in the case of the young knockouts. Figure 5b shows the results of the quantification of CTGF (connective tissue growth factor) immunohistochemistry, which is a downstream mediator of TGF-β1 and an early fibrosis marker. It was evident that age was positively correlated with increased expression of CTGF in both the wild-type and *Col8*-ko groups of both sexes, as already observed in TGF-β1 ELISA. The multifactorial ANOVA analysis confirmed this observation, demonstrating a significant effect of age (*p* = 0.037). The comparison between the wild-type and knockout groups revealed implied effects. In both the young and old male groups, the *Col8*-ko resulted in the increased expression of CTGF compared to the corresponding *Col8*-wt. As a result, the old male knockout group expressed the most CTGF of all the male groups. However, when examining the opposite sex, this knockout effect was only observed in the younger female groups, which showed a slight increase in CTGF expression.

## 3. Discussion

COL8 is a collagen that has not been extensively researched, and its pathological potential, not to mention its mechanisms of action, are not yet fully understood. Pathomechanisms in various organ systems have been linked to COL8, including the vascular system [12], the eye [25] and the kidney, whereas it has primarily been studied in the context of diabetic nephropathy (DN) [9,19,26]. The intent of this study was to investigate whether COL8 contributes to the pathophysiology of the aging kidney, since at least the structural changes in the senility-associated nephropathy are similar to those in the diabetic kidney, including glomerulosclerosis, interstitial fibrosis, the arteriosclerosis of small blood vessels, the thickening of basement membranes and the atrophy of the renal tubule. In addition, human urinary proteome assays and the cDNA microarrays of kidneys showed high similarity between aging and chronic kidney disease, such as DN [27,28].

The first indication that COL8 may play a role in age-associated mechanisms in the kidney was our finding of an age-dependent regulation that interacts with sex. This age-dependent expression was more pronounced in male kidneys and for the alpha 2 chain of COL8 than for COL8A1 and female kidneys.

Our studies describe, for the first time, an age- and sex-dependent *Col8* mRNA regulation in murine kidneys that was determined specifically for both chains. The data on *Col8* regulation in aging mouse kidneys is still very limited. That *Col8*-regulation is pathologically relevant and not the mere consequence of a fibrotic process is supported by the literature, which proves the role of COL8 in various pathologies.

Another important finding from our expression analyses of renal cortices from the different age groups of male and female wild-type mice was the mismatched regulation of *Col4a1*. Due to the distinct regulation of COL8 and COL4, it is likely that they have different functions, although they are probably both involved in the structure of the ECM in a similar manner, as type IV collagen is a network-forming collagen [29], like COL8.

Our comparative studies of kidneys from elderly mice that are wild-type for COL8 with those that do not express COL8 (*Col8*-ko) confirmed our hypothesis that COL8 could play a role in the pathophysiology of the aging kidney. In male wild-types, COL8 and damage increased with age, but to a greater extent due to knockout. This suggests that it is not COL8 itself but other mechanisms that are responsible for the age-induced renal changes, which are apparently inhibited by COL8. This pattern of structural changes in *Col8*-wt and *Col8*-ko males is reflected in the renal function (in the form of tubular function assessed by urinary cystatin C). Tubular dysfunction can be understood as a consequence of, or at least temporally downstream of, structural changes. Serum creatinine serves as a marker of renal function in clinical practice for the detection of renal dysfunction because it is freely filtered across the glomerulus and is neither reabsorbed nor metabolized by the kidney (although tubular secretion does occur) [30]. However, serum creatinine has important limitations; it can take 24–36 h to rise after a definite renal insult, it may overestimate renal function as a result of secretion in the proximal tubule, and it can increase following the administration of medications that inhibit tubular secretion despite no change in renal function. In addition, creatinine is distributed in total body water and measured as a concentration and may, therefore, be affected by variations in volume status [30]. Furthermore, changes in muscle mass and protein metabolism significantly affect serum creatinine levels. Therefore, the concentration of creatinine measured in the serum represents the balance between creatinine production and creatinine clearance [30]. Moreover, in humans, urinary protein profiles are valuable for diagnosing kidney diseases. In contrast to humans, mouse urine contains major urinary proteins (MUPs) resulting in a physiological proteinuria. The high level of MUPs in urine may mask the presence or absence of urinary proteins of similar sizes in the analyses of urinary proteins, making the determination of proteinuria in mice difficult [31]. The measurement of cystatin C in urine (UcysC) has been shown to be a suitable and non-invasive method to detect renal pathologies; increased UcysC concentrations allow for the accurate detection of tubular dysfunction among pure and mixed nephropathies [22]. It was further found that the ratio of UcysC to Ucrea (UcysC/Ucrea) is also a good index of the state of cystatin C reabsorption in the proximal tubules [32]. Since it is controversial whether and when urine biomarkers should be normalized against urine creatinine (Ucrea) concentrations, it is recommended to report both absolute and normalized values [33]. Variations in the urinary creatinine excretion rate between individuals occur because of differences in age, sex, race, diurnal creatinine production, physical activity, diet, emotional stress, muscle mass and disease state [33]. For example, patients with chronic kidney disease have a lower urinary creatinine excretion rate than those without chronic kidney disease. Urinary creatinine excretion rate may also decline as kidney disease progresses due to an increase in the extrarenal degradation of creatinine [33]. In fact, our data, showing that *Col8*-ko resulted in significantly increased urinary cystatin C excretion and an increased cystatin C/creatinine ratio in elderly mice, are not only true for males but could also be observed with the same intensity in females. In our model, we were, therefore, unable to detect any sex differences in renal function, as measured here by damage to the tubule cells. However, our analyses of the age-induced fibrotic changes in the extracellular matrix in the mesangium and interstitium showed a different picture in the female animals than in the males described above. There was no significant increase in the glomerular or tubulointerstitial matrix in old wild-type females compared to young females. This is consistent with the age-related sex differences in the aging kidney that have long been known. In human, sexual dimorphism is evident both in structural renal changes and in the impairment of renal function, with the male sex usually having a worse prognosis in both aspects [34]. Also, in rodents, a clear difference in age-related kidney damage between sexes is apparent, with aging females of most strains being substantially protected from such damage [34].

As a result of the *Col8*-ko in females, glomerular and interstitial fibrosis was exacerbated to the same extent as in male knockouts, indicating that the significant sex difference in the older wild-type groups was abolished by the knockout. These findings support the notion that COL8 has a renoprotective function in preventing tissue fibrosis in the aging kidney, with a pronounced effect in the female sex. On the other hand, studies in the context of diabetic nephropathy and cardiac fibrosis suggest that *Col8*-ko may be protective against fibrosis and disease progression, rather than collagen VIII itself [19,35]. This is not necessarily contradictory, as it is possible that fibrosis mechanisms are different in old age than in young age. In this regard, it is important to note that the mentioned studies of diabetic nephropathy and cardiac fibrosis were conducted using young animals, and the role of COL8 may vary depending on age. This also fits well with our findings in young *Col8*-ko vs. *Col8*-wt animals, where glomerular as well as interstitial fibrosis were reduced when COL8 was missing.

An abnormal extracellular matrix is a histologic feature of kidney aging and disease. However, a comprehensive molecular basis for an altered matrix is not well understood [36]. It is well-known from diabetic kidney diseases that an excessive presence of collagen IV is a good marker of fibrotic events, including the thickening of basement membranes [37]. In contrast, in the aging kidney, there is not an increase but a decrease in basement membrane components, which is consistent with our data, especially in wild-type females. In a study using a mouse model of kidney disease, Randles and colleagues observed that matrix composition varied with age. They found that basement membrane components (laminins, type IV collagen, and type XVIII collagen) were reduced, while interstitial matrix proteins (collagens I, III, VI, and XV; fibrinogens; and nephronectin) were increased [36]. A very surprising finding, shown here for the first time, was that the absence of COL8 in knockout animals led to a dramatic reduction in the essential component of basement membranes, namely COL4, throughout the kidney. It has been shown that a strong reduction in COL4 expression, as is already the case in the young *Col8*-ko animals, is a hallmark of severe damage to the kidney architecture [36].

It has also been found that patients with advanced chronic kidney disease have less COL4 than the comparison group, contributing to the development of kidney disease [37]. Therefore, reduced presence may indicate a disturbed composition and organization of the renal extracellular matrix. It is widely acknowledged that collagens serve not only as structural proteins but also as active participants in tissue turnover. Collagens interact with, and influence, each other, maintaining a homeostatic state. This is particularly important in processes that involve structural changes, such as aging [36]. An imbalance in collagen turnover favors the increased deposition of collagens and is also closely linked to fibrosis progression [37]. In kidney diseases, there may be a disrupted turnover of matrix proteins, leading to the degradation and the absence of collagen IV, and resulting in the overexpression of other collagens. Therefore, it is possible that fibrosis increases in old knockout mice, as shown in the Masson staining, while collagen IV decreases in young knockout mice, as shown in the IHC staining.

Considering the limited knowledge on the exact composition of the basement membrane of peritubular capillaries and the specific changes in the adventitia (such as thickening in sclerosis/perivascular fibrosis) in renal fibrosis, it is highly probable that there is an interaction between COL8 and COL4. The renal arteries are composed of the following three distinct compartments: the intima, which includes the subendothelial basal membrane and subendothelial interstitial ECM; the media, which contains smooth muscle cells and predominantly elastic fibers; and the adventitia, which consists of the interstitial type of collagen-rich ECM. A thorough examination of the vascular extracellular matrix in the kidney is currently lacking, despite the likelihood of its similarity to vessels in other organs, which has been reviewed elsewhere [24]. COL8 was identified as an endothelial collagen, while COL4 is a crucial component of the basement membrane. As both collagens can appear in different compartments of vascular architecture and COL8 can interact with basalmembranes [12], it is hypothesized that the lack of COL8 due to the knockout has a direct impact on the formation of the basement membrane and supporting matrix, including COL4, in young individuals. The elevated levels of COL4 in aged knockout mice, as observed by IHC staining, may indicate overcompensation due to age-related fibrosis.

Since renal dysfunction can usually be a direct consequence of the structural reorganization of the renal tissue, renal dysfunction is expected even in young *Col8*-ko animals due to the dramatic loss of COL4. And, indeed, the UcysC/Ucrea ratio in these animals shows an extremely large distribution in males and a biologically relevant increase in females.

To recapitulate, our data suggest that the pathological stimuli of aging (regardless of the nature) are more kidney damaging in males than in females, as evidenced by the fact that old wild-type males show more glomerular and interstitial fibrosis than wild-type females of the same age. Furthermore, the upregulation of COL8 in males could be a compensatory mechanism to protect the kidney—the protective potential of COL8 has been demonstrated in knockout mice. The lower pathological pressure of aging in females, in turn, does not require the compensatory upregulation of COL8 expression, although the absence of COL8 and perhaps COL4 in females is causative for the full expression of pathophysiological changes in the kidney, which is apparently suppressed by other mechanisms in the wild type.

The imbalance of the matrix architecture due to the downregulation of COL4 provides a relatively conclusive explanation for the observed effects in *Col8*-ko. Further studies focusing on the profibrotic TGF-β1-CTGF axis should provide further insights into the underlying mechanisms. CTGF is a recognized component of the fibrotic cascade in various organs, including the kidney. As a downstream mediator, CTGF is associated with TGF-β1, which also functions as a central part of the profibrotic cascade in kidney disease [38]. A significant age effect was evident in the renal TGF-β1 level, which was also observed with CTGF, showing that old age in wild-type mice of both sexes is associated with an increasing expression of both cytokines in the renal tissue. This finding was to be expected and is consistent with the literature [39]. However, there were sex differences in the *Col8*-ko groups; while, in males, the knockout resulted in an increased expression of TGF-β1 and CTGF, in females, there were no differences compared to the wild-type. The studies suggest that both markers are part of a common COL8-dependent fibrotic cascade in the male sex, in which this profibrotic pathways may be more activated due to the age and *Col8*-ko, indicating again a potentially protective role of COL8. Since the investigated parameter accumulations of the extracellular matrix and tubular dysfunction in the knockout do not differ in both sexes, one or more other mechanisms seem to be predominant in the female kidney. Although our observations do not allow for direct mechanistic conclusions, there is evidence that *Col8*-ko influences TGF-β1-induced pathways. Upon activation through TGF-β1, *Col8*-ko mouse-derived mesangial cells showed the activation of mitogen-activated protein (MAP) kinases and Akt pathways rather than Smad3, compared to wild-type cells [26]. TGF-β accordingly contributes to the progression of epithelial-mesenchymal transition (EMT) by activating MAPKs. MAPKs play a pivotal role in proliferation, differentiation, and inflammation, and can induce renal fibrosis [40]. The upregulation of TGF-β1 and the possible shift to alternative, pro-fibrotic TGF-β1 pathways resulting from the knockout in the older male group may explain the increased renal fibrosis observed in the same group and supports the idea that COL8 may have a protective effect on the kidneys of older individuals.

Sex hormones as possible direct contributors could play an important role in this respect. In part, the finding that females are substantially protected against the age-dependent decrease in renal function that occurs in males reflects the renoprotective effects of estrogens. On the other hand, estrogen has multiple actions, not all of which are beneficial. In addition, the low androgen level in women might be protective against a decline in renal function, because the presence of androgens is associated with increased renal damage and dysfunction, but animal and clinical data on possible adverse effects of androgens are controversial. Androgens also have multiple actions, one of which—aromatization to estrogen—is likely to be protective [34].

Since there were also clear differences in the age- and sex-dependent expression of both COL8 chains, it is possible that the individual chains have different importance in the COL8 effects shown. By using the *Col8a1*/*a2* double knockout, it was not possible, in this study, to investigate whether one of the two chains plays a greater pathological role in the aging kidney than the other. Further studies with only one knockout would be useful and would provide deeper mechanistic insights.

Of course, the results observed here in this animal study cannot simply be extrapolated to humans, as this study is of a descriptive nature. There are already human data that indicate a regulation of *Col8* in the course of aging [36], and in the human DN we have already been able to show the clinical relevance of COL8 [41], which also suggests a potential role of COL8 in the aging human kidney. Further follow-up studies are required to demonstrate the clinical relevance and pathophysiological significance of COL8 in the aging human kidney. One possible approach, in addition to classical COL8 expression studies from kidney biopsies, could be non-invasive urine measurements of COL8 excretion via extracellular vesicles—a quasi COL8 biomarker study.

## 4. Materials and Methods

### 4.1. Animals

*Col8α1*−/−/*Col8α2*−/− knockout mice (*Col8*-ko) were generated, as previously described [25]. Age-matched wild-type mice (*Col8*-wt) from the same breeding cohort were used as the control. Mice were crossed back for at least 10 generations into the C57BLKS/J background. The mice were kept in a pathogen-free animal house with 12 h light/dark cycle on standard chow and water ad libitum. All experiments were performed in accordance with the guidelines of the German Animal Welfare Act (§7) and were done in accordance with the German Animal Protection Law. At the end of the experiment, mice were placed in metabolic cages (Tecniplast, Buguggiate, Italy) to collect the 24 h urine. Mouse kidneys were removed and one kidney per mouse was fixed in 10% phosphate-buffered formalin and embedded in paraffin for histological and immunohistochemical studies. The remaining kidney was snap-frozen and used for the isolation of total RNA. For the assessment of the wild-type expression pattern of *Col8a1*, *Col8a2* and *Col4a1* over the age (n = 5–10 per group), male and female mice were divided into very young (vy; 6–11 weeks), young (y; 12–24 weeks), middle-aged (ma; 24–40 weeks), middle-aged/old (ma/o; 41–72 weeks) and old (o; 72–100 weeks). To study *Col8* genotype-based differences in young (y; mean 21.4 weeks (12–25 weeks) vs. old (o; mean 85.1 weeks (73–93 weeks)) (n = 5 per group), *Col8*-ko mice were also included.

### 4.2. RNA Extraction, cDNA Synthesis, and Real-Time PCR (RT-PCR)

Total RNA was isolated from kidney cortex homogenates (homogenizer SpeedMill P12 (Analytik Jena Bio Solutions, Jena, Germany)) using the RNeasy Lipid Tissue Mini Kit (Qiagen, Hilden, Germany), according to the manufacturer’s instruction. Possible DNA contaminations were eliminated using the RNase-Free DNase Set (Qiagen, Hilden, Germany). In total, 1 μg of total RNA was reverse transcribed into cDNA using the Reverse Transcription System (Promega, Madison, WI, USA). The determination of gene-expression levels was performed by semi-quantitative real-time PCR by use of the LightCycler-FastStart DNA Master SYBR Green 1 (Roche Diagnostics, Mannheim, Germany) and a thermocycler (qTower, Analytik Jena, Jena, Germany). RT-PCRs were carried out with sense and antisense primers at a concentration of 0.25 µM each (purchased from TIB Molbiol, Berlin, Germany). The program was completed after 35 cycles. The sequences and annealing temperatures of all primer pairs are listed in Table 1. The relative expression ratio was quantified by the ΔΔCT method, values were normalized to the average of Hypoxanthine phospho-ribosyltransferase 1 (*Hrpt1*) and Glyceraldehyde-3-Phosphate Dehydrogenase (*Gapdh*), and the young male group was assigned as an arbitrary value of 1.

### 4.3. Immunohistochemical Staining (IHC)

In preparation for immunohistochemical staining, 3 µm paraffin kidney sections from all animals were dried, dewaxed and rehydrated. A heat-mediated antigen retrieval procedure in AR9 Buffer (pH 9.0) or AR6 Buffer (pH 6.0) (Akoya Biosciences, Marlborough, MA, USA) (except for connective tissue growth factor (CTGF)) was performed.

For IHC staining, the blocking of endogenous peroxidase was achieved by incubation with 3% H_2_O_2_ (Carl Roth GmbH & Co.KG, Karlsruhe, Germany) for 10 min at room temperature. Following blocking with Roti-Block, the sections were incubated with primary antibodies overnight at 4 °C. The following primary antibodies were used: rabbit polyclonal anti-CTGF antibody (Abcam, Cambridge, UK) and rabbit polyclonal anti-Collagen IV (COL4A1) antibody (Abcam, Cambridge, UK). After the incubation of the sections with peroxidase-labeled goat anti-rabbit IgG antibody for 1 h (SeraCare, Milford, MA, USA), di-aminobenzidine (DAB) (DAB-peroxidase substrate kit; Vector Laboratories, Burlingame, CA, USA) was used as a chromogen. The tissue sections were counterstained with hemalum.

All slides were mounted with ProLong™ Diamond Antifade Mountant.

Whole-slide images were taken of the sections using the multispectral scanner Vectra Polaris™ (Akoya Biosciences, Malborough, MA, USA) and visualized by the software Phenochart 1.1.0 (Akoya Biosciences, Marlborough, MA, USA). The immunostained area was quantified in a pixel-based approach using an open-source image analysis platform (QuPath, version 0.4.4; https://qupath.github.io/; accessed on 27 September 2023) with a set threshold [46]. The same threshold was used for all sections. All sections were controlled for staining quality, threshold, and annotation correctness. Values are expressed as a percentage of the areas of interest.

### 4.4. Histochemical Analysis of Tissue Fibrosis (Masson’s Trichrome Staining)

As described above, the paraffin-embedded tissues were sliced at 3 µm thickness, dewaxed, and rehydrated. Masson’s trichrome staining was then performed using the standard protocol. Whole-slide images were taken of the sections using the multispectral scanner Vectra Polaris™ (Akoya Biosciences, Malborough, MA, USA) and visualized by the software Phenochart 1.1.0 (Akoya Biosciences, Marlborough, MA, USA). For the quantification of renal glomerular and tubular fibrosis, Masson’s staining was visually assessed for collagen blue staining in the glomeruli and in the tubulointerstitium, and quantified with a score of 0–5 (0 = no blue staining; 1 = <10%, 2 = 10–20%, 3 = 20–40%, 4 = 40–60%, and 5 = >60%).

### 4.5. Urine Analysis

ELISA analyses for Cystatin C (R&D Systems, Minneapolis, MN, USA), and Creatinine (Cayman Chemical, Ann Arbor, MI, USA) were then performed with murine 24 h urine by following the manufacturer’s instructions.

### 4.6. Analysis of Renal TGF-β1 Protein Expression

The isolation of the total protein from the cortices of mouse kidneys (n = 5 per group) was performed by homogenization with complete Lysis m buffer supplemented with protease inhibitors (both from Roche Diagnostics, Mannheim, Germany). The concentrations of activated TGF-β1 in the protein lysates of the renal tissue were quantitatively determined using the Mouse/Rat/Porcine/Canine TGF-β1 ELISA according to the manufacturer’s instructions (Quantikine, R&D Systems, Wiesbaden, Germany).

### 4.7. Statistical Analyses

The data are shown as box/whisker-dot plots, drawn using SPSS statistics (IBM company, Armonk, NY, USA). Each dot represents one mouse. The boxes’ boundaries mark the 25th percentile in the bottom and the 75th percentile on the top, while the whiskers, drawn below and above the box, span from the 10th to the 90th percentile. The line in the center of the box indicates the median of values.

For statistical calculation, SPSS statistics were used (IBM company, Armonk, NY, USA). The main effects and interactions were assessed by using a two-way or three-way analysis of variance (ANOVA) with age (young vs. old) and sex (male vs. female) as the two and age, sex, and genotype (wild-type vs. *Col8*-knockout) as the three factors, respectively. Interaction determines whether the one main effect depends on the level of the other main effect. For the intergroup comparison of multiple groups, the Kruskall–Wallis H/Mann–Whitney U test was used to test the statistical significance between two independent groups.

A *p*-value of ≤0.05 was considered statistically significant. The *p*-values from ≥0.05 to ≤0.1 were considered biologically relevant.

## 5. Conclusions

This descriptive work shows, for the first time, that the *Col8* genotype can influence the effect of age on fibrotic/structural and functional changes in the murine kidneys. In addition, *Col8* knockout dramatically alters the matrix signature, which may provide an explanation for the genotype–age interaction shown. Definite sex differences were detectable in the age-dependent regulation of the expression of both COL8 chains and the development of tissue fibrosis. Although there were no striking COL8-dependent sex differences in tubule function, it was shown that in males but not in females COL8 can have an influence on the TGF-β1-CTGF axis. Further studies are needed to better understand the underlying pathomechanisms, but these results provide initial clues to the age-related remodeling of the renal architecture and the associated decline in renal function.

## Figures and Tables

**Figure 1 ijms-25-04805-f001:**
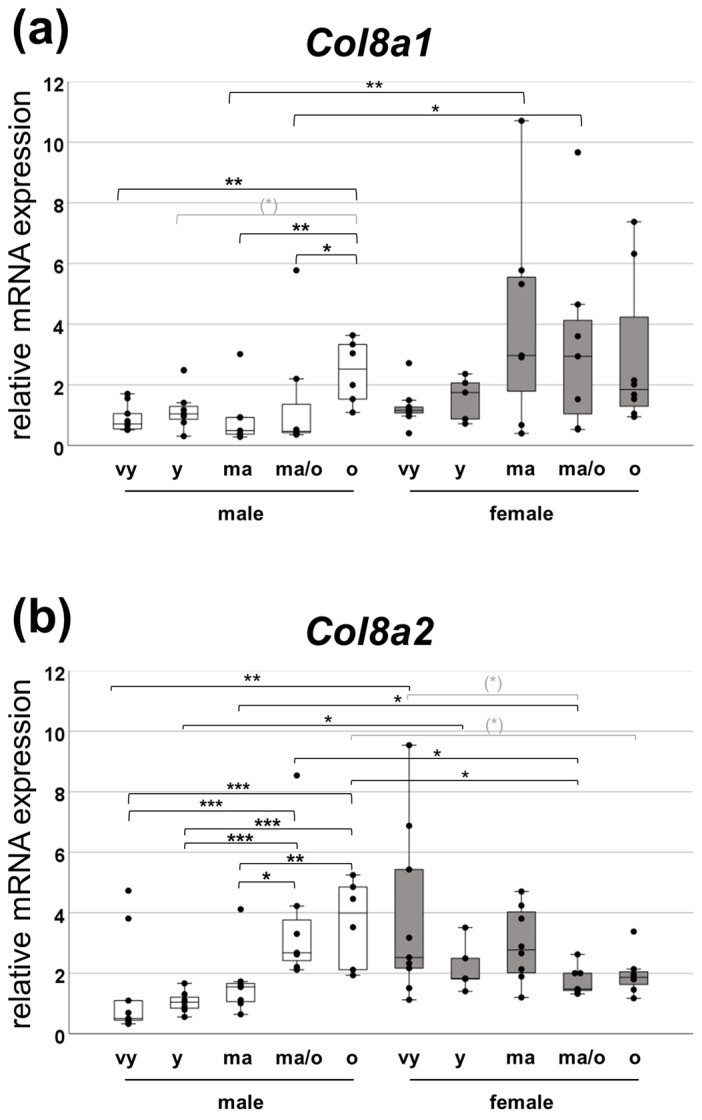
Expression of *Col8a1*, *Col8a2* and basement membrane collagen *Col4a1* in mice of different ages. Semi-quantitative mRNA expression of (**a**) *Col8a1*, (**b**) *Col8a2* and (**c**) *Col4a1* in the renal cortex assessed by RT-PCR. Values were normalized to the expression of the housekeeping genes *Hprt1* and *Gapdh*. Young (y) male group was assigned as an arbitrary value of 1. Transcript levels of all animals were normalized to the mean value of the young (y) male group. (n = 7). Significance and interaction were calculated by two-way ANOVA with sex and age as two factors. Sex, age: main effect of sex/age, respectively. Sex*age: interaction. Effects and interactions of *Col8a1* assessment: factor sex (*p* = 0.008), factor age (*p* = 0.081), Effects and interactions of *Col8a2* assessment: sex*age interaction (*p* < 0.001). Mann–Whitney U tests were performed for intergroup significance (shown above bars). Statistically significant *p*-values: * = ≤ 0.05; ** = ≤ 0.01; *** = ≤ 0.001. The symbol (*) in grey means biologically relevant with *p* ≤ 0.1. vy: very young (6–11 weeks); y: young (12–24 weeks); ma: middle-aged (24–40 weeks); ma/o: middle-aged/old (41–72 weeks); and o: old (72–100 weeks).

**Figure 2 ijms-25-04805-f002:**
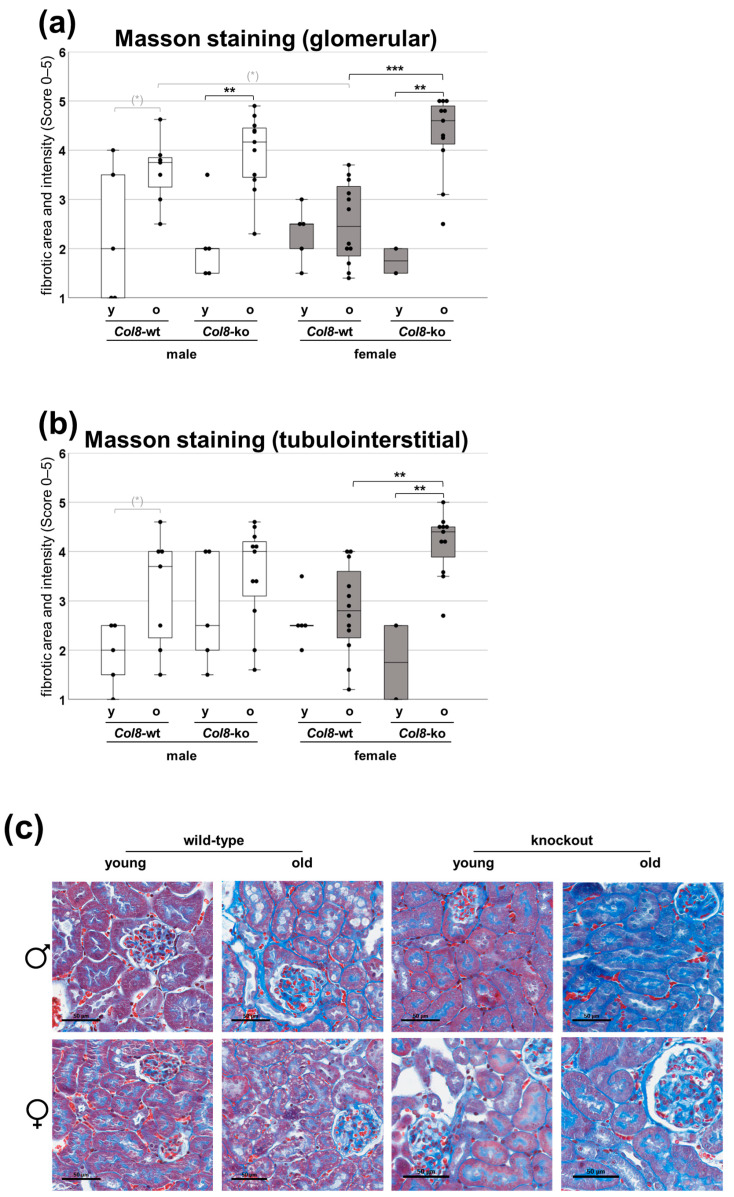
Pathological assessment of renal fibrosis depending on the factors *Col8* genotype, age and sex. (**a**,**b**) Quantification of glomerular and tubular fibrotic area (assessed by Masson’s trichrome staining). (**c**) Representative pictures of Masson’s trichrome staining (n = 5/7/5/11/5/12/2/11; number of animals per group from left to right). Significance and interaction were calculated by three-way ANOVA with genotype (*Col8* wild-type (wt) vs. *Col8* knockout (ko)), sex (male vs. female), and age (young vs. old) as the three factors. Effects and interactions of glomerular assessment: factor age (*p* < 0.001), genotype*age interaction (*p* = 0.007), and genotype*sex*age interaction (*p* = 0.091). Effects and interactions of tubular assessment: factor age (*p* < 0.001) and genotype*sex*age interaction (*p* = 0.017). Mann–Whitney U tests were performed for intergroup significance (shown above bars). Statistically significant *p*-values: ** = ≤ 0.01; and *** = ≤ 0.001. The symbol (*) in grey means biologically relevant with *p* ≤ 0.1.

**Figure 3 ijms-25-04805-f003:**
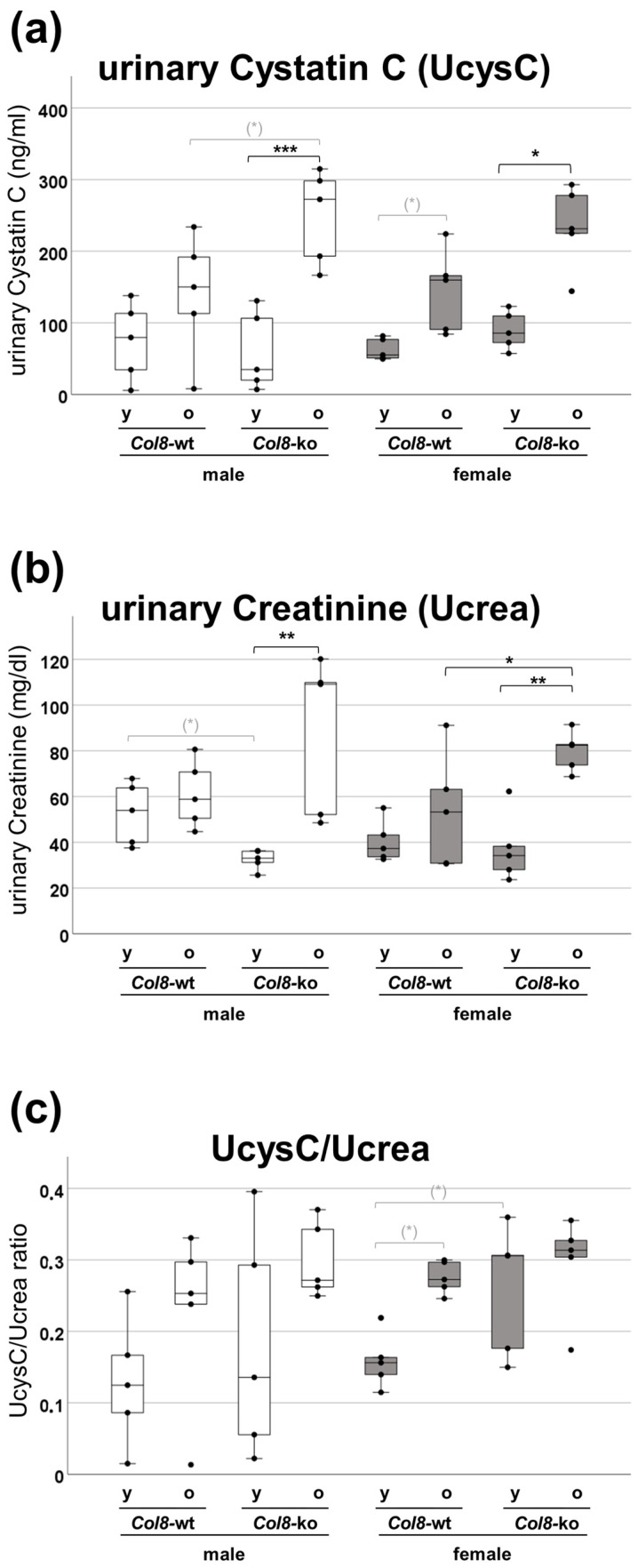
Role of COL8 in age-induced tubular dysfunction in mice of both sexes. (**a**) Protein analysis of urines with cystatin C ELISA. (**b**) Protein analysis of urines with creatinine ELISA. (**c**) Urinary cystatin C/urinary creatinine ratio (shown in µg/mg per ml) (n = 5). Significance and interaction were calculated by three-way ANOVA with genotype (*Col8* wild-type (wt) vs. *Col8* knockout (ko)), sex (male vs. female), and age (young vs. old) as the three factors. Effects and interactions of UcysC assessment: factor genotype (*p* = 0.006), factor age (*p* < 0.001), and genotype*age interaction (*p* = 0.014). Effects and interactions of Ucrea assessment: factor age (*p* < 0.001) and genotype*age interaction (*p* = 0.002). Effects of UcysC/Ucrea assessment: factor genotype (*p* = 0.043) and factor age (*p* = 0.003). Mann–Whitney U tests were performed for intergroup significance (shown above bars). Statistically significant *p*-values: * = ≤ 0.05; ** = ≤ 0.01; and *** = ≤ 0.001. The symbol (*) in grey means biologically relevant with *p* ≤ 0.1.

**Figure 4 ijms-25-04805-f004:**
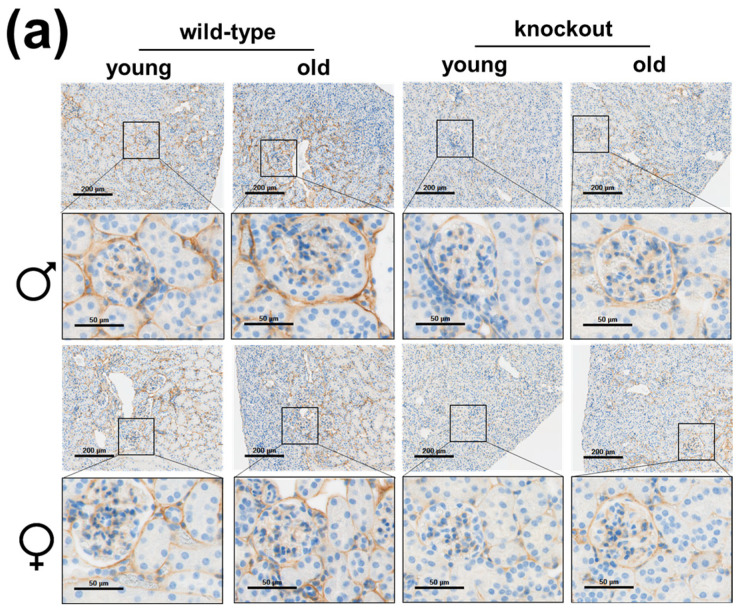
Renal COL4A1 protein expression depending on age, sex and *Col8* genotype. (**a**) Representative immunohistochemistry images of COL4A1 in the kidney sections of the indicated groups; brown staining indicates protein expression (blue: nuclei). (**b**) Quantification of collagen IV (COL4A1) protein expression (assessed by IHC). (n = 5). Significance and interaction were calculated by three-way ANOVA with genotype (*Col8* wild-type (wt) vs. *Col8* knockout (ko)), sex (male vs. female), and age (young vs. old) as the three factors. Effects and interactions of COL4A1 assessment: factor genotype (*p* < 0.001) and genotype*age interaction (*p* = 0.008). Mann–Whitney U tests were performed for intergroup significance (shown above bars). Statistically significant *p*-values: ** = ≤ 0.01; and *** = ≤ 0.001. The symbol (*) in grey means biologically relevant with *p* ≤ 0.1.

**Figure 5 ijms-25-04805-f005:**
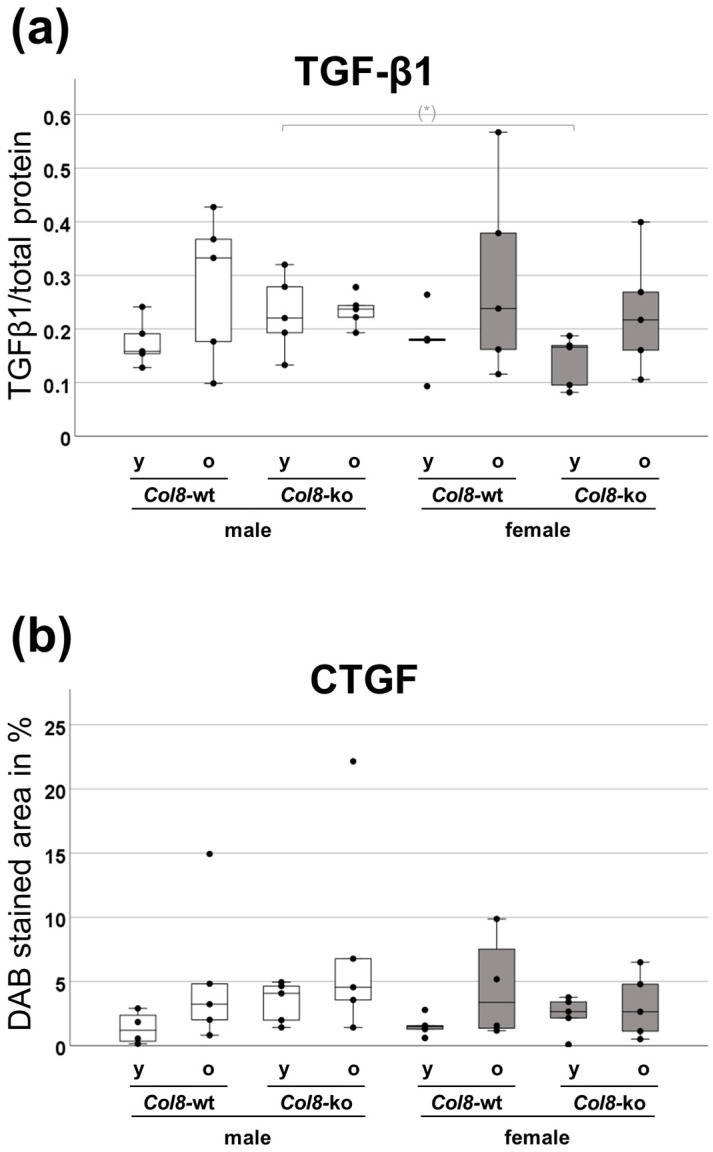
Influence of *Col8* genotype, sex and age on the expression of the profibrotic marker in mouse kidneys: (**a**) TGF-β1 ELISA of renal cortical tissue; (n = 5) (**b**) Quantification of connective tissue growth factor (CTGF) protein expression (assessed by IHC). (n = 5) Significance and interaction were calculated by three-way ANOVA with genotype (*Col8* wild-type (wt) vs. *Col8* knockout (ko)), sex (male vs. female), and age (young vs. old) as the three factors. Effects and interactions of TGF-β1 assessment: factor age (*p* = 0.018). Effects of CTGF assessment: factor age (*p* = 0.037). Mann–Whitney U tests were performed for intergroup significance (shown above bars). The symbol (*) in grey means biologically relevant with *p* ≤ 0.1.

**Table 1 ijms-25-04805-t001:** Gene primers and their respective annealing temperature.

Gene	Sense and Antisense Primers	Tann
*Col8a1*	5′-AGAGTGCACCCAGCCCCAGT-3′	66 °C [42]
	5′-TGGGTGGCACAGCCATCACATTT-3′	
*Col8a2*	5′-CCTGCAGGCTCTGCCTGTCC-3′	53 °C [42]
	5′-CACTCTTGGCCCACACCCCA-3′	
*Col4a1*	5′-TAGGTGTCAGCAATTAGGCAGG-3′	63 °C [43]
	5′-TCACTTCAAGCATAGTGGTCCG-3′	
*Hprt1*	5′-CAGATTCAACTTGCGCTCATC-3′	59 °C [44]
	5′-TGGATACAGGCCAGACTTTGTT-3′	
*Gapdh*	5′-CAATGACCCCTTCATTGACC-3′	59 °C [45]
	5′-TGGACTCCACGACGTACTCA-3′	

*Col8a1* = collagen type VIII alpha 1; *Col8a2* = collagen type VIII alpha 2; *Col4a1* = collagen type IV alpha 1; *Hprt1* = hypoxanthine phosphoribosyl transferase; *Gapdh* = glyceraldehyde 3-phosphate dehydrogenase.

## Data Availability

Data are contained within the article.

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
