# Peer review of "The Role of Collagen VIII in the Aging Mouse Kidney"

_ijms, 2024, doi:10.3390/ijms25094805_

Round 1
Reviewer 1 Report
Comments and Suggestions for Authors
In the present manuscript authors aimed to identify the role played by non-fibrillar collagen VIII in the progression of fibrosis in a mouse model of aging kidney. They evaluated some pro-fibrotic markers such as renal function by urinary cystatin C and they conclude that COLVIII can influence renal fibrosis and kidney function in the aging kidney. My major concern is related to the lack of mechanisms that can support the described results. Moreover description of results in a limited number of mice, no evidence on human tissues is reported to translate the obtained results and give them a clinical significance. Moreover authors should better try to explore the significant differences observed between males and females. Do they evaluate hormones in animals? Regarding kidney function, did authors evaluate proteinuria in mice? As minor point, the title should be revised and limited to the results obtained in mice
Comments on the Quality of English LanguageMinor editing of English language is required
Author Response
In the present manuscript authors aimed to identify the role played by non-fibrillar collagen VIII in the progression of fibrosis in a mouse model of aging kidney. They evaluated some pro-fibrotic markers such as renal function by urinary cystatin C and they conclude that COLVIII can influence renal fibrosis and kidney function in the aging kidney.
- My major concern is related to the lack of mechanisms that can support the described results.
We agree with the reviewer that this work is descriptive. Further investigations are of course necessary to understand the underlying mechanisms. We have provided explanatory approaches and hypotheses on the one hand with the described imbalance of the matrix architecture due to the strong reduction of COL4 expression and on the other hand with the influence of COL8 on TGF-β1 and CTGF signaling. In the discussion, we have further elaborated on the approach regarding TGF-β1.
- Moreover description of results in a limited number of mice, no evidence on human tissues is reported to translate the obtained results and give them a clinical significance.
It is certainly true that this is a preclinical study and the results observed cannot simply be translated to humans. Follow-up studies are of course necessary. However, there are already human data that indicate a regulation of COL8 in the course of ageing and in the human DN we have already been able to show the clinical relevance of COL8 in kidney disease. We have taken up this point in the discussion.
- Moreover authors should better try to explore the significant differences observed between males and females. Do they evaluate hormones in animals?
The reviewer is absolutely right: the sex differences are an important point. In this study, this was a very interesting secondary finding, in addition to the role of COL8 expression, which we definitely want to shed more light on in the future. In this context, we will also analyze the sex hormones. Future studies on the observed sex differences must be significantly expanded beyond the mere determination of sex hormone levels. The pure endpoint determinations of the sex hormones alone will probably not have much significance - follow-up controls would have to be carried out. This in turn is difficult, as relatively large amounts of serum are required, at least for ELISA tests, which are difficult to carry out in living mice. Furthermore, it would make sense to carry out stimulation experiments with β-estradiol and DHT at the in vitro level and castration experiments in vivo in order to test the direct influence of the sex hormones.
- Regarding kidney function, did authors evaluate proteinuria in mice?
We did not investigate proteinuria. Since this study focused mainly on tubulointerstitial and less on glomerular changes and as it has been described that severe renal damage can also exist without albuminuria occurring, we focused on the determination of tubular dysfunction via urinary cystatin C.
- As minor point, the title should be revised and limited to the results obtained in mice
We have changed the title:“ The role of Collagen VIII in the ageing mouse kidney“.
- Minor editing of English language is required
The manuscript has been revised with regard to the English language.
Reviewer 2 Report
Comments and Suggestions for Authors
Vo et al. investigated the effects of sex and age on COL8 expression in mouse kidneys. Their main findings are that COL8 is regulated in an age- and sex-dependent manner and that COL8 expression influences the severity of age-induced renal fibrosis and function. The experiments seem to be carefully designed and executed. The authors used more methods to investigate the potential role of COL8 in kidney fibrosis with aging. The text is generally well-written and easy to understand. This Reviewer has only minor comments.
1. Please provide the genetic background of the mice in the Methods section (e.g., C57BL/6).
2. Please give the cycle number of the qPCR reactions in the Methods section.
3. The authors mention that they used urine collected for 24 h. Therefore, the animals should be placed into metabolic cages to collect urine. Please describe the method by which you collected urine.
4. If the authors collected blood at the termination of the experiment, it would be helpful to show the serum creatinine values and calculate the creatinine clearance. Did the authors measure urine albumin or total protein?
5. On the Figures, some of the significance signs are written in grey and others in black. What is the meaning of the grey stars?
6. Figures: The content of the table inserts could be moved into the figure legend or the text of the MS (only the significantly changed values). Fig. 1-3: Vertical orientation of the panels would be better. Generally, panels have different font sizes in the Figures. Please try to use no more than 2 different font sizes in each Figure.
Author Response
Vo et al. investigated the effects of sex and age on COL8 expression in mouse kidneys. Their main findings are that COL8 is regulated in an age- and sex-dependent manner and that COL8 expression influences the severity of age-induced renal fibrosis and function. The experiments seem to be carefully designed and executed. The authors used more methods to investigate the potential role of COL8 in kidney fibrosis with aging. The text is generally well-written and easy to understand. This Reviewer has only minor comments.
- Please provide the genetic background of the mice in the Methods section (e.g., C57BL/6).
Done.
- Please give the cycle number of the qPCR reactions in the Methods section.
Done.
- The authors mention that they used urine collected for 24 h. Therefore, the animals should be placed into metabolic cages to collect urine. Please describe the method by which you collected urine.
This is correct, in order to be able to collect 24-hour urine, the animals had to be placed in metabolic cages. We have added this information to the methods section. Thank you for pointing this out.
- If the authors collected blood at the termination of the experiment, it would be helpful to show the serum creatinine values and calculate the creatinine clearance. Did the authors measure urine albumin or total protein?
Unfortunately, we do not have enough blood/serum from these animals to calculate the creatinine clearance. We also did not investigate proteinuria. Since this study focused mainly on tubulointerstitial and less on glomerular changes and as it has been described that renal damage can also exist without albuminuria occurring, we focused on the determination of tubular dysfunction via urinary cystatin C.
- On the Figures, some of the significance signs are written in grey and others in black. What is the meaning of the grey stars?
In our study, we differentiate between statistical significance (P < 0.05) and biological relevance (P 0.1 - 0.05). To illustrate this visually, we have placed the asterisks in brackets and written them in gray if we mean biological relevance. The figure legends now explain what "gray" asterisks mean. Thank you for pointing out that this was unclear.
- Figures: The content of the table inserts could be moved into the figure legend or the text of the MS (only the significantly changed values). Fig. 1-3: Vertical orientation of the panels would be better. Generally, panels have different font sizes in the Figures. Please try to use no more than 2 different font sizes in each Figure.
Thank you very much for the comment. We have removed the tables from the figures and moved the significances to the figure legends. They are also named in the main text. The different font sizes were caused by subsequent resizing after insertion into the Word template. The panels are now arranged vertically, as recommended.